neuroscience

microglia, pain, neuropathy

**Author for correspondence:**
Steven J. West
e-mail: stevenjonwest@gmail.com

# Microglia: sculptors of neuropathic pain?

Harry Ward[1] and Steven J. West[2]

[1]Nuffield Department of Clinical Neurosciences, University of Oxford, Oxford, UK
[2]Sainsbury Wellcome Centre, University College London, 25 Howland St, London WC1E 6BT, UK

SJW, 0000-0002-2413-0999

Neuropathic pain presents a huge societal and individual burden. The limited efficacy of current analgesics, diagnostic markers and clinical trial outcome measures arises from an incomplete understanding of the underlying mechanisms. A large and growing body of evidence has established the important role of microglia in the onset and possible maintenance of neuropathic pain, and these cells may represent an important target for future therapy. Microglial research has further revealed their important role in structural remodelling of the nervous system. In this review, we aim to explore the evidence for microglia in sculpting nervous system structure and function, as well as their important role in neuropathic pain, and finally integrate these studies to synthesize a new model for microglia in somatosensory circuit remodelling, composed of six key and inter-related mechanisms. Summarizing the mechanisms through which microglia modulate nervous system structure and function helps to frame a better understanding of neuropathic pain, and provide a clear roadmap for future research.

## 1. Introduction

Neuropathic pain is defined as 'pain caused by a lesion or disease of the somatosensory nervous system' [1]. It affects up to 10% of the population (reviewed in [2]). These patients have poorer health and greater disability [3], reduced quality of relationships [4] and productivity, which costs, combined with other pain conditions, up to $1 trillion per year [5] and €14 446 per patient in EU countries [6]. Less than 50% of current analgesics provide even 50% pain relief [7] and there are significant problems in drug interactions, convenience, adherence and little effect on physical and emotional functioning. This is probably due to reliance on redundant, non-specific targets and assessment by traditional randomized controlled trials with inappropriate outcome measures that do not consider biopsychosocial factors (reviewed in [8,9]).

New therapeutic targets for neuropathic pain management arise from continued understanding of the underlying biology.

Indeed, microglia represent an important cell type initiating [10] and perhaps maintaining [11,12] neuropathic pain. Microglial research has also begun to unravel important mechanisms for these cells in sculpting neural circuitry across the central nervous system (CNS). In this review, we draw on and review important discoveries in basic microglial function, from which a growing number of studies highlight the importance of these cells for diverse aspects of neuronal remodelling and the shaping of nervous system function.

We further explore the latest research into microglial mechanisms in driving neuropathic pain states [13–16], and critically analyse important insights relating structural changes of neuronal circuitry to the neuropathic phenotype (reviewed in [17,18]). Finally, we synthesize a new model for microglia in somatosensory circuit remodelling, composed of six key and inter-related mechanisms, that define the contributions of these mechanisms to the neuropathic pain state.

# 2. Microglia as sculptors

Arising from the yolk sac at embryonic day 7.5, microglia migrate into the brain by E10.5 with minimal contribution from haematopoiesis [19], giving rise to a population maintained by colony-stimulating factor (CSF1)-dependent proliferation [20] and nestin$^+$ cell differentiation [21]. Microglia show significant heterogeneity in space [22] and time [23–26], and are dependent on environmental cues. They are thought to have distinct structural phenotypes: ramified, primed, active and amoeboid [27], which mediate different effects on neuronal structure, excitability and survival.

Microglia express a host of metabotropic and ionotropic receptors (reviewed in [28]). Some drive the formation of different phenotypes, such as lipopolysaccharide (LPS) receptor toll-like receptor 4 (TLR4) which mediates an inflammatory phenotype. Others are involved in mediating functions of certain phenotypes, such as migration, secretion and cell-surface binding. These include the CSF1 receptor, the fractalkine receptor CX3C chemokine receptor 1 (CX3CR1) and purinergic receptors (reviewed in [29]). CSF1 and its receptor CSF1R play an important role in microglial development, as it has been shown to stimulate microglial enzymatic activity *in vitro* [30], and although microglial development appears to progress in the absence of CSF1 *in vivo*, the ablation of the CSF1R, which is also stimulated by interleukin-34 (IL-34) [31], significantly impairs microglial migration to the brain during development [19]. The stimulation of CSF1R appears to be essential for the maintenance and retention of microglia in adult brain, as inhibition of this receptor results in the almost complete loss of the microglial population [21]. Migration of microglia can be directed by purinergic receptor P2Y12 in response to adenosine triphosphate (ATP) and adenosine diphosphate (ADP) [32] which is released from active neurons [33]. The inflammatory phenotype is thought to be mediated in part by CX3CR1 activity which drives the production of inflammatory cytokines interleukin-1$\beta$ (IL-1$\beta$) and tumour necrosis factor (TNF-$\alpha$) [34], whereas the release of brain-derived neurotrophic factor (BDNF) from microglia is driven by purinergic receptors P2X4 which drive changes in neuronal excitability [35]. In this way, these receptors allow communication with neurons and the environment to tailor their activation state and function.

Microglia are a glial cell type found throughout the CNS, commonly referred to as resident macrophages. While they were initially thought to be solely immune cells responding to damage, it is becoming increasingly clear that they have a more active role in neuronal signalling (reviewed in [36]). More current theories include the 'quad-partite synapse' where a synapse is not formed solely between two neurons but is also under the modulation of two types of glial cells: astrocytes and microglia [37]. This modulation can be functional through alterations in excitability or structural through alterations in the number of synapses and neurons in the circuit. Remodelling of neuron structure by microglia contributes to neurodegenerative [38–40] and psychiatric disease states [41,42], and here we explore its potential role in pain memory after nerve injury that drives the chronicity of neuropathic pain.

## 2.1. Microglia as surveillants

The study of microglial surveillance *in vivo* was revolutionized by the pioneering skull-thinning technique developed by Nimmerjahn *et al.* [43].[1] Live imaging of microglia using enhanced green fluorescent protein (EGFP) under the control of the CX3CR1 gene promoter revealed dynamic surveillants of synapses even in

---

[1]Seminal paper showing that activation of microglia not necessary for function and suggesting that microglia play a role in normal physiology and not just pathology.

resting state, with highly motile processes that contact neurons and glia [43] driven by activity-dependent purinergic signalling [44,45], disproving fixed-tissue reports that cannot gauge dynamism [46]. This surveillance has functional implications, as microglial contact duration on synapses correlates with decreased neuronal $Ca^{2+}$ activity [47] and prolonged contact of ischaemic neuronal terminals correlates with the disappearance of pre-synaptic boutons [48]. Furthermore, P2Y12-knock-out mice show defects in migration, preventing surveillance and synapse targeting during ocular dominance plasticity [49]. Monocular occlusion in the ocular dominance model promotes contact with neurons in the primary visual area of cortex (V1), spine turnover and an increased perimeter around synapses in the occluded region [50], suggestive of axon and extracellular matrix remodelling. Importantly, microglial surveillance is altered by neurotransmitter stimulation [51,52], which indicates that microglial surveillance is modified based on local neuronal activity and possibly direct neuronal-microglial signalling. Microglial surveillance may decline with age due to senescence which results in smaller and slower processes, slower migration and a more sustained response to injury [53].

## 2.2. Microglia actively prune synapses

The correlation of microglial contact and spine density suggests the involvement of microglia in synaptic maintenance. The key studies investigating the underlying mechanisms are outlined in table 1. A role for microglia in synaptic pruning, defined as the removal of synaptic elements, classically but not exclusively by engulfment of synaptic terminals through phagocytosis (figure 1a), was first seen by co-localization of synaptic markers and microglia in the developing hippocampus [65].[2] However, three-dimensional reconstruction, pHrodo-dextran labelling [37] and time-lapse imaging [66] show phagocytosis of only pre-synaptic elements, although studies are mostly restricted to the developing tectum and hippocampus. Nonetheless, in CX3CR1 knock-out mice, microglial density is reduced while spine density and connectivity are increased [65]. Interestingly, in this model, there are more immature synapses in the CA1 region of the hippocampus with more release sites, resulting in seizures [65]; this may indicate that microglia usually target these immature synapses over more well-established ones. However, neurons and immune cells may also be affected in this model as they also express CX3CR1 [67,68]. In fact, synaptic withdrawal of axotomized neurons is normal in the absence of microglial proliferation [69] and retrograde nitric oxide signalling mediates neurite retraction [70] suggesting that microglia are not always required for this process.

Insights into the targeting mechanism came from mice with knock-outs of complement factor 3 (C3) or the C3 receptor (CR3), part of the innate immune system. In these animals, there was a lack of activity-dependent refinement of the synapses in the developing retino-geniculate system with increased synapse density and unopposed pre-synaptic elements [37,71]. Importantly, microglia are the only cell type to express CR3 in this system, suggesting that they must be mediating this selective synaptic pruning process [37,71]. This gave rise to the complement tagging hypothesis where C3 and complement factor 1q (C1q) tag synapses that are then eliminated by microglia (figure 1a). However, in a mouse model of Alzheimer's disease with progranulin-deficient microglia, concurrent knock-out of C1q did not fully prevent the selective pruning process [68]. This model still showed increased lysosomal activity in microglia, co-localization of microglia with parvalbumin$^+$ interneurons, and reduced vesicular GABA transporter$^+$ (VGAT$^+$) puncta which represent inhibitory synapses [72]. Therefore, while complement-mediated synaptic pruning relies on microglia, other interactions may allow specific targeting of inhibitory neurons. Furthermore, it has not been shown whether complement tagging allows enhanced phagocytosis of less-active inputs [37]. Tagging specificity for these less-active synapses could arise from activity-dependent exosome release, as evidence *in vitro* shows that incubation of microglia with exosomes enhanced complement-dependent removal of degenerating neurites in a cell-type specific manner [73], potentially by providing cell-type specific ligands for competition between synapses. The environment also impacts on microglia–neuron interactions; sialic acid on the glycocalyx inhibits complement binding and phagocytosis and its absence during oxidative stress may promote remodelling [74]. Activity-dependent signalling in neurons may also modulate pruning; inhibition of nuclear $Ca^{2+}$ signalling in spinal neurons in inflammatory pain models prevents maintenance of mechanical hypersensitivity and prevents a reduction in C1q levels [75]. C1q application reduces spine density and mechanical hypersensitivity which is reversed by C1q knock-down with short hairpin RNA (shRNA) [46]. This suggests that increased neuronal activity reduces C1q expression and pruning, thereby preventing synaptic pruning and precipitating

[2]Seminal paper introducing the concept of synaptic pruning by microglia.

**Table 1.** Key studies investigating mechanisms of neural circuit sculpting by microglia.

| mechanism | brain region | model | observations | reference |
|---|---|---|---|---|
| pruning | developing retinogeniculate system | mouse | labelled microglia show dose apposition to synapses and contain pre-synaptic elements | [37] |
| | | | pruning targets less-active inputs after pharmacological manipulation of activity | |
| | | | pruning of synapses was significantly reduced in CR3 knock-out | |
| | | | microglial engulfment of synapses significantly impaired in CR3 and C3 knock-outs | |
| | motor cortical projection neurons | male SD rats | activation of microglia with LPS reduced GAD$^+$ punctae | [54] |
| | | | microelectrode array in motor cortex showed increased power of $\gamma$-band suggestive of synchronicity | |
| | | | showed increased levels of BDNF, FGF2, Bcl-2 which promote survival | |
| | | | pharmacological inhibition of microglia with minocycline prevented effect on GAD$^+$ punctae and reduced neuronal survival | |
| | organotypic hippocampal slices; cultured hippocampal neurons | mouse | depletion of microglia with clodronate liposomes increased frequency of excitatory postsynaptic currents suggesting increased synaptic density | [55] |
| | | | co-incubation of microglia with cultured neurons reduced synapse density and levels of adhesion molecules synCAM-1 and protocadherin | |
| | hippocampal CA1 neurons and dorsal horn | adult male mice | after spared nerve injury, spine density in CA1 reduced but increased in dorsal horn with impaired long-term potentiation (LTP) in hippocampus but increased LTP in the dorsal horn, this was mirrored by decreased BDNF levels in hippocampus and increased levels in the dorsal horn, TNF$\alpha$ increased in both regions | [56] |
| | | | pharmacological inhibition and genetic ablation of microglia prevented upregulation of TNF$\alpha$ and changes in spine density in both regions | |
| | | | TNFR1 knock-outs showed no change in spine density after spared nerve injury | |

(Continued.)

**5**

**Table 1.** (*Continued.*)

| mechanism | brain region | model | observations | reference |
|---|---|---|---|---|
| synaptogenesis | motor cortex layer V pyramidal neurons | mouse | tamoxifen-induced Cre system to drive diphtheria toxin receptor expression in $CX3CR1^+$ cells allowing diphtheria toxin-inducible depletion of microglia depletion resulted in reduced spine formation and reducing learning removal of BDNF from microglia broadly phenocopied the original depletion experiment with the exception of the novel object task | [57] |
| | cultured hippocampal neurons | rat | applied microglia to cultured hippocampal neurons without direct contact resulting in increased synaptic density, showed high levels of IL-10 recombinant IL-10 application increased synaptic density knock-down of IL-10 prevented synaptic formation, as did LPS pre-treatment | [58] |
| | dorsal root ganglion | adult female SD rats | application of artemin, a glial derived neurotrophic factor (GDNF), after lumbar dorsal root injury failed to regenerate large diameter myelinated afferents but induced regeneration of nociceptive smaller diameter calcitonin gene-related peptide ($CGRP^+$) axons, thereby enhancing recovery of nociceptive behaviour, and also increased isolectin B4 ($IB4^+$) axon regeneration NGF showed differential targeting of regenerating axons with increased density of deeper laminae and no effect on $IB4^+$ axons | [59] |
| | somatosensory cortex | mouse | *in vivo* multiphoton imaging of developing cortex in Iba1-EGFP mice, labelled neurons with RFP, used GCaMP6m to visualize calcium transients and lifeact-mCherry to visualize actin showed microglial contact induced calcium transients, F-actin accumulation and filopodial formation which was reduced by pharmacological inhibition and genetic ablation of microglia | [60] |

(*Continued.*)

**Table 1.** (*Continued.*)

| mechanism | brain region | model | observations | reference |
|---|---|---|---|---|
| death | Purkinje cells developing cerebellum | mouse | showed microglia with Purkinje cell marker inclusions surrounded by apoptotic Purkinje cells | [61] |
| | | | elimination of microglia with clodronate liposomes increased number of Purkinje cells | |
| | | | superoxide and hydrogen peroxide scavengers and NADPH oxidase inhibitors reduced Purkinje cell death | |
| | neuron cultures | mouse | LPS-treated microglia induced neurite beading and death which was also seen by TNFα, this was prevented by MNDAR inhibition and glutamine starvation suggesting excitotoxicity | [62] |
| | | | neutralizing antibodies against TNFR1 prevented this increase in glutamate levels and also glutaminase expression in microglia | |
| | | | gap junction inhibitors prevented glutamate accumulation suggesting a role for microglial glutamate release | |
| | cultures, optic nerve, post-mortem brain tissue | SD rats, mouse, human | knock-outs of CSF1, which lack microglia, showed a lack of A1 astrocyte induction | [63] |
| | | | LPS-treated microglia induce A1 astrocytes as do IL-1α, TNF, C1q | |
| | | | co-culturing A1 astrocytes with retinal ganglion cells results in almost 100% cell death | |
| | | | after optic nerve crush, neutralizing antibodies against IL-1α, TNF, C1q to prevent A1 astrocyte formation prevented retinal ganglion cell death; this was also seen in triple knock-out IL-1α/TNF/C1q mice which lack A1 astrocytes | |
| | | | A1 astrocytes are found in post-mortem tissue from patients with Alzheimer's, Huntington's, Parkinson's, amyotrophic lateral sclerosis and multiple sclerosis | |

(*Continued.*)

**Table 1.** (*Continued.*)

| mechanism | brain region | model | observations | reference |
|---|---|---|---|---|
| survival | layer V cortical neurons | mouse | microglial staining showed accumulation on post-natal tracts in layer V which then dispersed at P14 | [64] |
| | | | pharmacological inhibition of microglia with minocycline and genetic ablation increased TUNEL$^+$ (apoptotic) cells in layer V | |
| | | | *in vitro* microglia decreased cell death in a transwell system, detected IGF1 in medium | |
| | | | inhibitor of IGF1R phosphorylation and knock-down of IGF1 attenuated survival of these layer V neurons *in vivo*, and IGF1 was able to rescue cell death in minocycline-treated animals | |
| | motor cortical projection neurons | male SD rats | activation of microglia with LPS reduced GAD$^+$ punctae | [54] |
| | | | microelectrode array in motor cortex showed increased power of $\gamma$-band suggestive of synchronicity | |
| | | | showed increased levels of BDNF, FGF2, Bcl-2 which promote survival | |
| | | | pharmacological inhibition of microglia with minocycline prevented effect on GAD$^+$ punctae and reduced neuronal survival | |

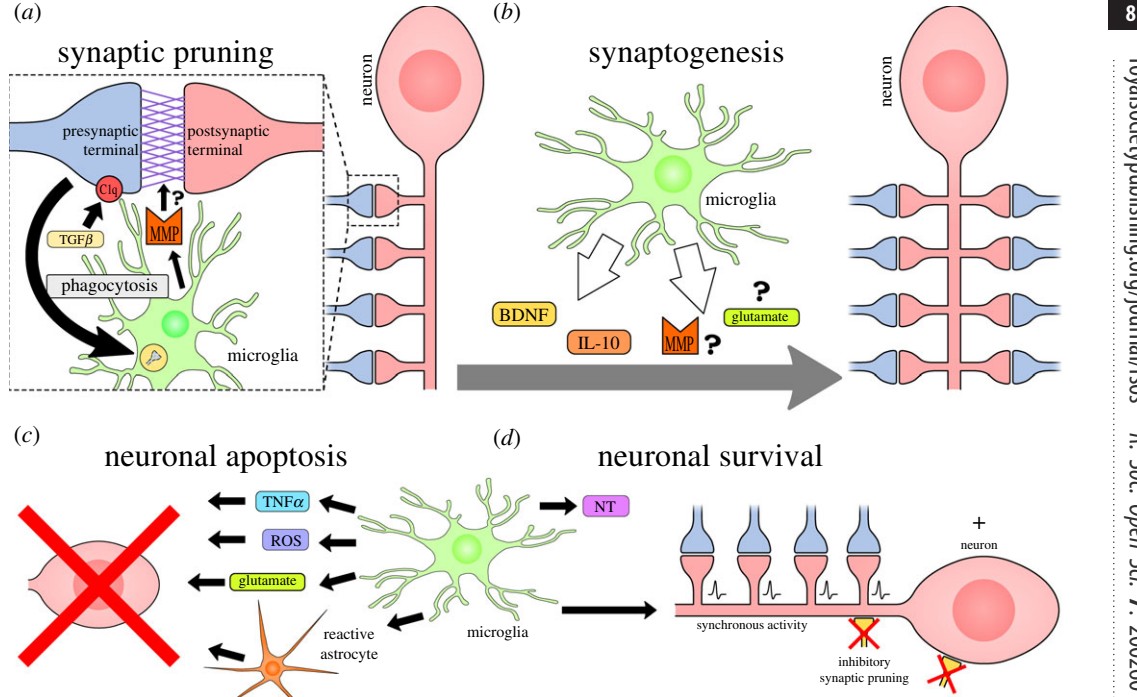

**Figure 1.** Microglial sculpting mechanisms. Microglia constantly survey their local environment, and have been shown to contribute to the following structural changes in the central nervous system. (*a*) Synaptic pruning can be driven by C1q tagging of synapses, which promotes the phagocytosis of presynaptic terminals. TGF$\beta$ promotes C1q expression for this tagging process. Synaptic removal may rely on protease activity by microglia to break down synaptic adhesion molecules. (*b*) Microglia secrete factors including BDNF, IL-10 and potentially MMP and glutamate to promote synaptogenesis. Adhesion molecule binding (NCAM, integrin) may also promote synapse formation. (*c*) Microglia drive neuronal cell death through secretion of ROS, TNF$\alpha$, glutamate and induction of reactive astrocytes. (*d*) Microglia promote neuronal survival through the provision of neurotrophic factors and growth factors (NT) and can promote synchronous synaptic activity through inhibitory synapse pruning which promotes survival pathways.

maladaptive hyperexcitability [75]. Surprisingly, a recent study suggested that neuronal C1q levels are insignificant as ablation of hippocampal neurons has little effect on overall C1q levels [76], although ablation may upregulate microglial C1q or perhaps neuron levels fluctuate and only become significant in pathology.

Microglia are also capable of complement-independent pruning to transiently displace inhibitory synapses [54], although the phenotype, environment and pathways promoting this mechanism are unknown. There is also evidence that trogocytosis, the transfer of membrane-associated proteins, between neurons and microglia in the developing hippocampus can occur in the absence of complement in a CR3 knock-out mouse [66]. However, this study lacked quantification of the knock-out and number of neurons and mice analysed. Trogocytosis appeared to be associated with synapse formation by guiding maturation [66], which probably requires specific environmental cues.

Another pruning mechanism could be the cleavage of trans-synaptic adhesion molecules and scaffolding proteins (figure 1*a*). Changes in synaptic protein levels seen in synaptic pruning could be explained by phagocytosis of the pre-synaptic terminals, although proteases are also known to be involved in synaptic remodelling as inhibition of metalloproteinases increases the levels of glutamatergic markers [77]. This may represent a synaptic breakdown pathway, whereby phagocytosis of the presynaptic terminal is accompanied with digestion of the synaptic cleft. Furthermore, optogenetic caspase-3 activation locally eliminates spines without inducing apoptosis suggesting that proteolysis is sufficient for pruning [78]. While microglia are well known to secrete proteases, such as metalloproteinase, further work is required to investigate the contribution of proteases to synaptic pruning by microglia in different contexts.

Cytokine signalling also appears to contribute synaptic remodelling. Although the causal mechanisms have not been defined, knock-out mice lacking TNF fail to show increased synapse elimination in response to an immunostimulant polyinosinic:polycytidylic acid, known as poly(I:C), while application of exogenous TNF$\alpha$ promotes pruning [67]. TNF$\alpha$ is known to play a role in synaptic plasticity such as long-term depression, which may manifest as shrinkage and elimination of spines [79,80]. Adenoviral

over-expression of IL-6 also alters the levels of synaptic markers. Interestingly, this over-expression results in reciprocal effects on excitatory versus inhibitory synapses with increased levels of glutamatergic but reduced levels of GABAergic markers and increased number, length and density of spines [81]; this reciprocal control could be key in pathological states characterized by hyperexcitability such as neuropathic pain. However, the mechanism leading to these changes is unknown. Direct administration of IL-6 *in vitro* shows that IL-6 is not itself synaptogenic [58] but it may activate immune cells capable of mediating this specific pruning of GABAergic synapses [59]. In summary, there is strong evidence that microglia can drive synaptic pruning and this is regulated in complex ways which are yet to be fully understood.

## 2.3. Microglia promote branching and synaptogenesis

Microglia have also been found to facilitate synapse formation, termed synaptogenesis, and therefore play a homeostatic role to guide the maturation of new synapses while also pruning inactive ones. A direct role for microglia in synaptogenesis was shown by Cre-driven ablation of CX3CR1$^+$ microglia in the hippocampus. Ablation resulted in reduced learning-dependent spine turnover, reduced freezing in a fear conditioning paradigm, and also reduced interest in novel objects, indicating abrogated learning and memory [57]. Removal of BDNF from microglia decreased spine formation and phenocopied fear response behaviour of the ablation model but not novel object behaviour [57], suggesting that BDNF release from microglia can contribute to synaptogenesis (figure 1*b*). Microglia-derived BDNF is also known to drive increased dendritic spine density in the hippocampus after nerve injury, which is prevented by microglial inhibitor minocycline or conditional ablation of microglia [56].[3] Other synaptogenic mechanisms may include IL-10 secretion by microglia (figure 1*b*). Recombinant IL-10 mimics synaptogenesis of hippocampal neurons seen in separated co-culture system with microglia, whereas shRNA knock-down of IL-10 levels prevents it [58]. The IL-10 receptor is expressed by cultured hippocampal neurons, and the knock-down of this receptor inhibited microglial synaptogenesis [58]. Importantly, despite non-specific release, neurotrophins are able to promote synaptogenesis of specific inputs, owing to differential expression of receptors, and this allows specific targeting of neuronal subsets by microglia [59]. However, reactive microglia appear to prevent the formation of synapses, as stimulation of co-cultures of microglia and hippocampal neurons with the immunostimulant LPS inhibits synaptogenesis [58], and the proinflammatory cytokine IL-1$\beta$ has been demonstrated to suppress BDNF-driven spine consolidation [82], which is likely to be driven through microglia. In summary, microglia are capable of aiding synaptogenesis, but may hinder synaptic development when they react to inflammatory stimuli.

Synaptogenesis by microglia may further be enhanced through direct microglial contact. Cultured hippocampal neurons in microfluidic chambers, which allowed only their axons to grow into a chamber containing a microglial cell line pre-primed with LPS, showed that synapsin1, an important marker of synapse formation, clustered around areas of contact with these pre-primed microglia [83]. However, this is a very artificial system and microglia cell lines are not representative of any *in vivo* phenotype [84].[4] More recently, *in vivo* multi-photon imaging of the developing somatosensory cortex showed microglial contact locally upregulates Ca$^{2+}$ levels and promotes actin polymerization to induce filopodial formation, the first step in synaptogenesis [60]. Although only 45% of contacts induce filopodia and filopodia form without microglial contact, there is a striking deficit when microglia are ablated [60]. Further studies are required to determine the mechanisms of recruitment and cell-surface interaction, and some suggested pathways are discussed below. It will be important to determine the molecular phenotype of microglia mediating this function to investigate its presence in pathology.

There are some cell-surface interactions that are already known to facilitate synaptogenesis which may be used by microglia. For example: homophilic γ-Protocadherin interactions [85], interactions between intercellular adhesion molecule 5 (ICAM-5), also called telencephalin, with lymphocyte function-associated antigen 1 (LFA-1) [86], or with integrin-β1 [87] which facilitates metalloproteinase-9-mediated spine modulation [88], interactions between thrombospondin and integrins [89], and neural cell adhesion molecule (NCAM) and N-cadherin homophilic and heterophilic interactions (reviewed in [90]). Indeed, NCAM2 clustering promotes filopodial formation through a cascade of

---

[3]One of the first papers showing a direct role for microglia in structural remodelling in neuropathic pain.

[4]One of the first papers to define the molecular signature of microglia.

$Ca^{2+}$, tyrosine kinase and $Ca^{2+}$/calmodulin kinase (CaMKII) signalling [91], which is also true of the immunoglobulin superfamily, cadherins and integrins (reviewed in [92]). Therefore, to bring about contact-dependent synaptogenesis, microglia would probably mediate cell-surface interactions through neural adhesion molecules which directly couple to F-actin via WAVE and actin-related proteins (Arp2/3) [93] or indirectly via intracellular $Ca^{2+}$ and kinases which modulate F-actin via kalirin and Rho GTPase [94]. These pathways converge to promote actin polymerization for filopodial protrusion which is stabilized by entry of microtubules to form a dendritic spine (reviewed in [95]).

Although it may seem that contact promoted synaptogenesis in previous studies [60,83], it may be explained by activity-targeted migration to specific regions of the neuron [96] and localized secretion of molecules (figure 1b). Local glutamate secretion by microglia may stimulate local N-methyl-D-aspartate receptor (NMDAR) $Ca^{2+}$ events [62,97,98] to modulate F-actin [94]. C1q secretion disinhibits the effects of myelin on neuritogenesis and therefore promotes synapse formation [99]. Metalloproteinase-9 promotes integrin-dependent actin polymerization for spine expansion [88] and tissue plasminogen activator facilitates formation of pre-synaptic varicosities [100] potentially by clearance of extracellular matrix or competing ligands for integrin receptors. Thus, microglia can draw on a plethora of possible mechanisms for synaptogenesis dependent on the environment and phenotype.

## 2.4. Microglia regulate neuron birth and longevity

Microglia are involved in maintaining homeostasis during the formation of new circuitry. This involves regulation of neurogenesis, cell death, and neuronal survival (figure 1c,d). Evidence supports the notion that microglia can promote neurogenesis through the secretion of inflammatory cytokines, such as IL-1$\beta$, IL-6, TNF$\alpha$ and interferon-gamma (IFN$\gamma$) [101], although it has also been demonstrated that pro-inflammatory microglia may inhibit neurogenesis [102], suggesting the microglial control of neurogenesis by a more nuanced phenotype. In the hippocampus, microglia can clear excess newborn cells by phagocytosis [103] so that only a small number mature and integrate into the neural circuitry. In P2Y12 knock-out mice, phagocytosis of apoptotic cells is reduced and hippocampal neurogenesis is downregulated [104], suggesting a role for these phagocytic microglia in the regulation of neurogenesis. In fact, phagocytic microglia secrete pro-neurogenic factors, such as vascular endothelial growth factor (VEGF) and fibroblast growth factor 2 (FGF2), that promote the formation of new neurons *in vitro* [104].

Similarly, during development, microglia drive cell death to control the number of neurons. In the developing cerebellum, reduced nicotinamide adenine dinucleotide phosphate (NADPH) oxidase activity in microglia produces reactive oxygen species (ROS) which drive neuronal death [61]. Further, TNF$\alpha$ secretion from somite macrophages is thought to drive cell death of developing motor neurons [105]. In pathology, these mechanisms may be hijacked. TNF$\alpha$ has also been shown to promote increased microglial glutamate release resulting in NMDAR-mediated excitotoxicity in Rett syndrome models [98]. Furthermore, cross-talk between glia can also play a role as microglia can induce A1 reactive astrocytes which are neurotoxic and implicated in neurodegenerative disease (figure 1c) [106]. In adult physiology, microglia are thought to respond to damage-associated molecular patterns (DAMPs) to activate microglia to a neurotoxic or a neurotrophic phenotype to allow clearance or repair respectively (reviewed in [63]).

Microglia are critical for neuronal survival and repair through secretion of neurotrophic factors and growth factors (figure 1d). Evidence for the role of microglia in providing these growth factors comes from depletion studies which show an increase in neuronal apoptosis. For example, apoptosis of developing callosal neurons was rescued by exogenous insulin growth factor 1 (IGF1) which was expressed by the microglia surrounding these neurons [107]. Microglia have also been shown to produce BDNF, glial-derived neurotrophic factor (GDNF), neurotrophin-3 (NT-3) and nerve growth factor (NGF) [64,108]. The provision of these neurotrophic factors is thought to promote survival and repair (reviewed in [63]). This is not always the case, however, as neutralizing antibodies against microglial-derived NGF in the developing retina prevented cell death [109], although this is likely to be a rare case. Microglia can also promote synchronous synaptic activity in cortical neurons through the displacement of inhibitory presynaptic terminals, resulting in neuroprotection through NMDAR signalling [54]; this contrasts excessive NMDAR activity which drives excitotoxicity [98]. Importantly, microglial phenotypes adapt to the environment to fulfil a neurotoxic or neuroprotective role (reviewed in [63]) and change over time; chronic LPS exposure causes a shift from a pro-inflammatory to a neuroprotective phenotype [110], consistent with temporal fluidity of different activation states.

# 3. Microglia contribute to neuropathic pain

Neuropathic pain is defined as pain arising from a lesion or disease to the somatosensory nervous system [1]. This definition covers a wide range of disorders, all of which may have contributions of different underlying mechanisms, that make neuropathic pain notoriously difficult to study (reviewed in [18]). Although a wide range of models have been developed that correspond with the diversity of underlying drivers of this condition, the most commonly employed model type is of localized injury to a peripheral nerve (reviewed in [111]). These encompass different injury types (crush, ligation, transection of the nerve) and different sites (most commonly the sciatic nerve or its branches, but also the oculomotor nerve and the saphenous nerve). The typical phenotype arising from these models includes thermal and mechanical hyperalgesia, allodynia (sensitivity to innocuous mechanical stimulation), and sensitivity to cold stimuli. Here we review the mechanisms that implicate microglia in the development and resulting phenotype of these neuropathic pain models.

## 3.1. Microglia maintain neuropathic pain symptoms

Proliferation of amoeboid microglia, termed microgliosis, has been shown in neuropathic pain models to occur in somatotopically specific regions of the dorsal horn [13] and higher regions such as the anterior cingulate cortex (ACC) [14] and frontal cortex [15]. There is some contention around supraspinal microgliosis [112], but this is probably owing to non-representative neuropathic pain models (reviewed in [113]), unreliable determinants of microglial activity, such as proliferation and morphology instead of molecular definitions, use of transgenic animals and *in vitro* protocols. Indeed, imaging of activated microglia with a positron emission tomography (PET) radioligand to a translocator protein (TSPO) in patients with chronic low back pain shows upregulation in higher regions, and that the levels of activated microglia correlated with pain and IL-6 levels [16].

An active role for these microglia in the pathogenesis of neuropathic pain has been shown through pharmacological and genetic methods to inhibit microglial function (reviewed in [28]). An early pharmacological intervention of intrathecal administration of minocycline to inhibit microglia was shown to prevented nociceptive behaviour [11,14,20]. However, there are currently no specific pharmacological agents to inhibit microglia; minocycline is neurotoxic [114], targets $Na^+$ channels [115], metalloproteinases [116] and ROS [117], leading to questions of off-target effects in these neuropathic pain protocols. It is also important to be able to target certain microglial phenotypes. Amoeboid microglia are thought to be more 'activated' [27], and a greater understanding of this state could provide specificity necessary for effective treatment.

Genetic methods have revealed a network of molecules and pathways that are thought to play a significant role in neuropathic pain biology (reviewed in [28]). Knock-out of CSF1 in dorsal root ganglion (DRG) neurons prevented microglial activation and hyperalgesia after nerve injury, and showed a significant decrease in expression of activated microglial genes such as CX3CR1, cathepsin-S and purinergic receptors P2X4R and P2Y12R [20]. Targeting these genes with knock-outs in neuropathic pain models has shown that both CX3CR1 and P2Y12 are required for microgliosis and the subsequent neuropathic pain phenotype [10,32], whereas mechanical hypersensitivity can still be elicited in P2X4 knock-out mice [20]. This is surprising given the critical role P2X4 activation plays in microglial BDNF secretion, which drives hyperexcitability in sensory neurons by promoting the expression of potassium-chloride co-transporter (KCC2) [35]. However, only the initiation of hyperalgesia was measured in P2X4 knock-out mice and not its maintenance; this would be in line with the concept of an inflammatory response driving the initial change in pain threshold, followed by structural remodelling to mediate long-term maintenance of hyperalgesia.

Greater temporal precision of the knock-out of these genes is required to unpick the role of activated microglia at different stages of neuropathic pain pathogenesis while also preventing potential effects of altered microglia on CNS development. Interestingly, microglial ablation early after the induction of a neuropathic pain model was shown to prevent hypersensitivity, whereas later ablation had little effect on hypersensitivity, leading the authors to conclude that microglia are only involved in initiation and not maintenance [68]. However, it may also imply that initial microglial-mediated remodelling could precipitate a vicious cycle of neuron-intrinsic maladaptive remodelling which does not require continued microglial involvement. Alternatively, structural remodelling may be mediated through a different microglial phenotype that does not express CX3CR1 or through activated microglia which were spared from ablation. This is consistent with an inducible knock-out of microglial BDNF in a neuropathic pain

model which showed initiation but not maintenance of hypersensitivity in response to nerve injury [11]. Furthermore, microglial inhibition [11,33,118,119] or TrkB blockade [12] after the initiation of neuropathic pain symptoms reverses allodynia, presumably before essential structural alterations to neurons, although these remain undefined. Indeed, inhibition of cytokines during the maintenance period did not affect excitability or prevent maintenance [12], further corroborating the concept that while the inflammatory phenotype initiates neuropathic pain symptoms, longer term alterations could be mediated by a distinct phenotype, which is consistent with the known fluidity of microglial phenotypes. Genetic studies specifically inhibiting different phenotypes during the initiation or the maintenance period would allow interrogation of this concept. However, an important consideration is the definition of the 'initiation' and 'maintenance' period; some models can take much longer for hyperalgesia and other effects of nerve injury to manifest (reviewed in [111]) and therefore a broad-brush comparison of timings could be misleading.

Importantly, studies now begin to characterize single-cell molecular phenotypes and genetic landscapes of 'activated' microglia [84,120,121]. High-throughput, non-biased computational algorithms trained on these data from different contexts could disambiguate these phenotypes to allow mechanistic insight and comparison between models and with patients. This shifts away from reliance on arbitrary structural definitions, which must be standardized to quantify a continuum of alterations [122], and the false dichotomy of M1/M2 phenotypes (reviewed in [123]), giving a nuanced spectrum of functional phenotypes.

## 3.2. Microglia contribute to structural changes in the somatosensory nervous system

Much of the research on microglial biology in neuropathic pain focuses on the secretion of molecules that alter neuronal excitability; however, this approach fails to take into account the role of microglia in modulating neuronal structure and survival. A change in structure and number of neurons may mediate pain memory that accounts for long-term symptoms seen in neuropathic pain without a continued stimulus. Importantly, studies have now started to investigate the role of microglia in structural alterations in the dorsal horn and hippocampus in a neuropathic pain model [56]. They show reciprocal control of spine density in these two regions, with an increase in the dorsal horn but a decrease in the hippocampus, mirrored by higher levels of BDNF in the dorsal horn compared to the hippocampus. In TNF receptor 1 (TNFR1) knock-out mice, this change in BDNF expression was not seen, and neither was the change in spine densities with nerve injury. This was also seen with minocycline administration and genetic ablation of microglia, which prevented the expression of TNFα and the spine density changes. Importantly, all of these models prevented mechanical allodynia after injury, suggesting that these structural alterations in spine density are required to mediate neuropathic pain symptoms [56].

Both BDNF and inflammatory cytokines, such as TNFα, were already known to be involved in neuropathic pain pathogenesis. BDNF alters cell-surface protein expression in dorsal horn neurons, driving KCC2 expression and promoting hyperexcitability [35]. Inflammatory cytokines, on the other hand, directly modulate synaptic currents, thereby promoting hyperexcitability, and activate downstream transcription factors such as cAMP response element-binding protein (CREB) [124] which may mediate more long-term alterations in structure [125]. In fact, neonatal rats, which curiously display a more anti-inflammatory profile of secreted cytokines after nerve injury, do not show neuropathic pain phenotypes [126], suggesting that the balance of anti-inflammatory and pro-inflammatory cytokines is important in neuropathic pain pathogenesis. Transcriptomic analysis of human microglia has also shown an increase in pro-inflammatory signalling with age [127]. Interestingly, this correlates with an increased prevalence of neuropathic pain in elderly patients [128], which may be related to this shift in the balance of anti-inflammatory and pro-inflammatory signalling in aged microglia. This is in keeping with their known opposing roles in trophic versus pruning activity on neuronal structure respectively [58,68]. Aged microglia also show a more sustained response to injury [53], which may promote the development of neuropathic pain in the elderly.

In CX3CR1 knock-out mice, one group have shown reduced microglial activation and improved axonal sprouting and synaptogenesis in the dorsal horn after spinal cord injury and showed increased survival and growth of descending inhibitory fibres [129],[5] suggesting a role for activated microglia in sculpting neuronal structure after nerve injury. This also presents a therapeutic avenue to promote reparative rather than inflammatory microglial phenotypes in neuropathic pain. However, the range of mechanisms employed by microglia to mediate these alterations in neuronal structure is unclear. Removal of a phospholipid

---

[5]Important paper showing that microglia mediate structural remodelling in neuropathic pain model.

scramblase (TMEM16F) on microglial membranes to prevent phagocytosis in one neuropathic pain model prevented mechanical hypersensitivity [130], suggesting a role for synapse engulfment and removal, although spine density was not investigated. It is, therefore, important to review the sculpting mechanisms employed by microglia in other regions of the CNS and then correlate this to neuropathic pain pathogenesis to provide a fuller picture of the potential underlying mechanisms.

# 4. Structural remodelling maintains neuropathic pain

Several mechanisms have been proposed that link structural alterations to neural circuitry with the neuropathic pain phenotype. Here we explore and critically assess the evidence for these mechanisms in neuropathic pain models.

## 4.1. Loss of inhibition

Genetic ablation of specific spinal interneurons results in spontaneous mechanical allodynia [131,132] and emphasizes the importance of GABAergic and glycinergic inhibitory gating of two pathways in allodynia: a polysynaptic A$\beta$ pathway via PKC$\gamma^+$ neurons and a direct pathway via vertical cells [133,134]. Death of inhibitory and top-down modulatory neurons may underlie allodynia in neuropathic pain. One study showed increased TUNEL staining, reduced glutamate decarboxylase 67 (GAD67) and inhibitory current amplitude after nerve injury which was prevented by caspase-3 blockade; importantly, this blockade also reduced hyperalgesia [135]. Some studies replicate these changes in inhibitory markers after nerve injury [136,137] whereas others do not [138,139]. Secreted factors from microglia also drive disinhibition [35,140–142] and alterations in synaptic proteins [143]. Interpretation of these studies may be complicated by apoptosis of descending fibres from higher regions [144] and neurogenesis potentially induced by injury [145].

## 4.2. Ectopic sprouting

While it was initially postulated that A$\beta$ primary afferent fibres, that convey innocuous mechanical information, may display ectopic sprouting to nociceptive-specific regions in dorsal horn, and this may drive allodynia by re-routing light touch stimuli to the pain pathway [146–149], this was later disproved as the labelling used was not specific to A$\beta$ neurons. Studies have since shown the presence of A$\beta$ nociceptors which were probably inadvertently assigned as mechanical A$\beta$ fibres in these initial experiments [150–154]. It now appears that allodynia is more likely caused by more subtle synaptic remodelling between light touch and nociceptive circuitry [150–154] However, denervation-induced nociceptive sprouting may precipitate extra-territorial hyperalgesia and allodynia [155].

## 4.3. Dendritic spine remodelling

Spinal contusion injury drives a Rac1-dependent increase in spine density, length and size with a shift towards the cell body to facilitate hypersensitivity [156],[6] [157] and hyperreflexia [158]. These alterations increase synaptic fidelity and reduce inhibitory efficacy [159]. Hypersensitivity induced by synaptogenesis is mediated by kalirin-7-dependent Rac1 activation [94] and LIM-domain kinase-dependent cofilin and CREB regulation [125]. Interestingly, cannabinoid receptors co-immunoprecipitate with WAVE1 (Wiskott–Aldrich syndrome protein-family verprolin homologous protein 1), a protein involved in regulating the actin cytoskeleton, prevent F-actin polymerization, and attenuate hypersensitivity after immunostimulant complete Freund's adjuvant (CFA) injection [160], presenting possible therapeutic avenues.

## 4.4. Supraspinal remodelling

MRI scans of patients with chronic pain show increased insula, ACC, pre-frontal and motor cortex thickness and activity that normalizes after surgery [161,162], suggesting that reversible structural and functional alterations correlate with chronic pain. A causative role for remodelling of connectivity is revealed by phantom limb pain [163] as anaesthesia only eliminates reorganization when pain is also

[6]Important paper introducing the concept where dendritic spine remodelling causes symptoms of neuropathic pain.

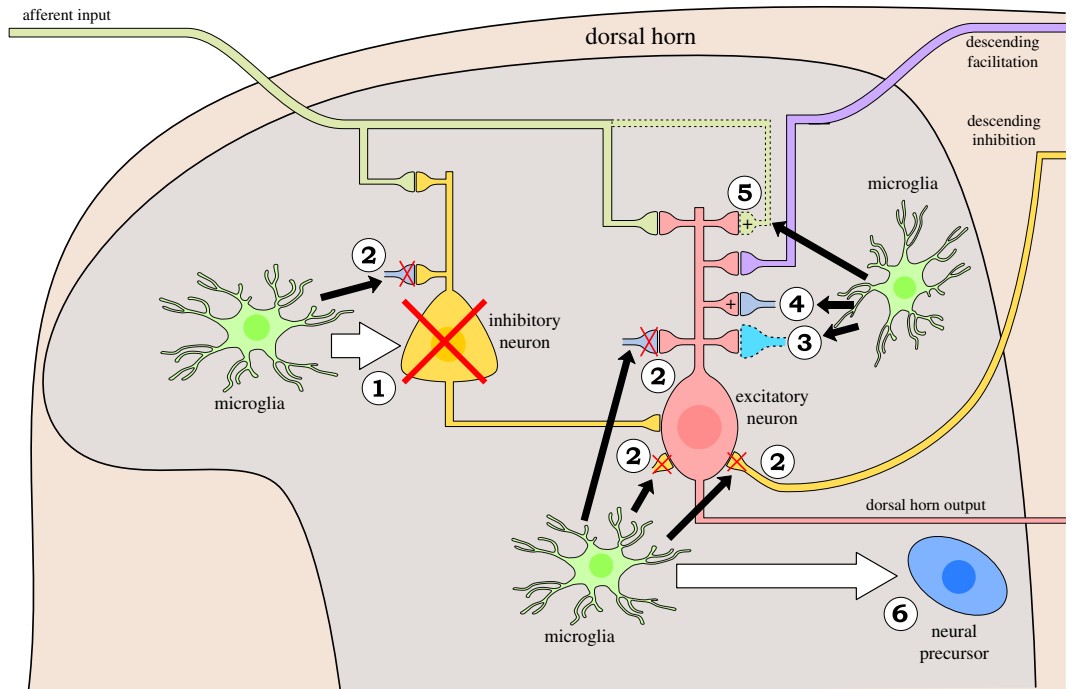

**Figure 2.** Schematic of microglial sculpting of pain pathway (1–6). Peripheral afferents synapse in the dorsal horn onto the pain pathway where they are modulated by interneurons, descending fibres from the brain (descending facilitatory and inhibitory influences), microglia and astrocytes after injury. We propose that microglia mediate structural remodelling of excitatory and inhibitory neurons of the pain pathway through six different mechanisms illustrated here. (1) Inhibitory cell death: death of interneurons (shown by a red cross) in the dorsal horn reduces inhibitory gating on pain transmission. (2) Synaptic pruning: pruning of synapses (indicated with red crosses) can alter the excitatory/inhibitory balance in the dorsal horn. (3) Synaptic remodelling: alterations in synaptic proteins may promote disinhibition of ascending fibres mediating pain transmission and mediate integration of immature neurons into dorsal horn circuitry. (4) Dendritic spine remodelling: increases in spine density promotes hyperexcitability. (5) Survival and ectopic sprouting: survival and sprouting of excitatory afferents and descending neurons may promote hyperexcitability and alterations in connectivity. (6) Neurogenesis: increased neurogenesis may remodel neural circuitry in the dorsal horn and mediate pain memory.

eliminated [164] and brain-machine interfaces inducing cortical plasticity modulate pain intensity [165]. Remodelling in neuropathic pain probably parallels associative learning where top-down modulation overrides bottom-up transmission [166].

These data on structural remodelling that drive hyperexcitability in neuropathic pain states are probably demonstrating different perspectives of the overarching phenomenon of circuit remodelling. Microglia are known in other contexts to reciprocally modulate inhibitory and excitatory neuron structure [56,72,167] as seen in neuropathic pain, and next we synthesize these different perspectives under one concept: microglial sculpting of the pain pathway.

## 5. Proposed mechanisms for microglial sculpting of the pain pathway

Finally, we propose a new framework linking microglial responses to peripheral nerve injury with the remodelling of somatosensory circuitry. From this model we derive several well-defined hypotheses where microglial activation may result in neuronal structure remodelling (figure 2).

Although current theories around microglial involvement in neuropathic pain are based on modulation of excitability through secreted factors (reviewed in [28]), the known sculpting ability of microglia and structural alterations after nerve injury begs the question of microglial involvement in this structural remodelling process. Although studies suggest that synaptic remodelling is restricted to the pre-natal period [37,85], spinal cord injury induces developmental genes [168,169] suggesting the reappearance of developmentally restricted mechanisms. Recent studies also suggest redundancy and sexual dimorphism in the contribution of microglia in neuropathic pain [11,68,170]. However, reports of sexually dimorphic microglial pathogenicity are inconsistent [11,170,171]. One study suggests that

adaptive immune cells can compensate for microglia in female but not male rats [11], but this is contested by recent evidence suggesting that adaptive immune cells do not infiltrate the dorsal horn after nerve injury [10]. Furthermore, the differences could be due to flaws in their methodology as ablation studies abrogate resting microglial function, inhibition studies require quantification and long follow-up times, and none of the studies defined the molecular phenotypes of microglia involved. These studies also only interrogated known microglial mechanisms in neuropathic pain and not those suggested here, and therefore further work is required to fully understand the role of sexual dimorphism in neuropathic pain.

Next, we discuss each of the potential underlying mechanisms for microglial sculpting of the somatosensory nervous system that contributes to the neuropathic pain phenotype.

## 5.1. Reciprocal control of inhibitory and excitatory neuron survival

There are several mechanisms by which microglia can mediate apoptosis: production of ROS, neurotrophins, pro-inflammatory cytokines or induction of reactive astrocytes (figure 2(1)). In a neuropathic pain model, microglia have been shown to release TNFα with effects on BDNF levels [56], which are known to modulate survival, although this is yet to be investigated in this context. Furthermore, ROS production has been correlated with spinal inhibitory neuron function [140]. Whether ROS production in microglia affects remodelling in neuropathic pain is an open question, which could be addressed with redox-sensitive GFP [172], and conditional inducible deletion of NADPH oxidase in microglia. Pro-inflammatory cytokines can promote neurotoxicity via death ligands such as Fas ligand and TNF-related apoptosis inducing ligand (TRAIL) [173,174], and this must be balanced by anti-inflammatory factors to prevent excessive apoptosis and promote resolution [175,176]. Interestingly, levels of microglial-derived TNFα; correlate with clinical parameters of fibromyalgia [177], suggesting a possible mechanism underlying fibromyalgia symptomatology. Furthermore, optogenetic activation of astrocytes modulates inhibitory neurons resulting in disinhibition, and induces a neuropathic pain phenotype.

Selective loss of inhibitory but not excitatory neurons [135–137], may be mediated by neurotrophins released from microglia [56], which drive survival and sprouting (figure 2(5)). This specificity in survival mechanisms, owing to differential neurotrophic receptor expression, has been previously reported [59]. These opposing but simultaneous actions of toxicity and regeneration have been shown in the spinal cord after an inflammatory stimulus [175] but require investigation in a neuropathic pain model. Further, although it appears that in peripheral nerve injury peripheral leucocytes fail to infiltrate the parenchyma [10], this is contested [68] and must be carefully scrutinized in different neuropathic pain models with PNS or CNS injury. A failure to infiltrate does not rule out their contribution to neuropathic pain pathogenesis, however, and this must also be addressed [178].

## 5.2. Pruning of inhibitory synapses

An important question is whether the decrease in inhibitory synaptic markers after nerve injury represents cell death or pruning of inhibitory synapses (figure 2(1) and (2)). C1q, proteases and TNFα are upregulated by microglia in neuropathic pain models [125,141,176,179] and these are all known to mediate synaptic pruning, as reviewed above, and therefore the machinery necessary for either process is present in neuropathic pain. However, in a neuropathic pain model with CX3CR1-deficient microglia, presumably incapable of mediating their normal function, there were more inhibitory synapses than wild-type and enhanced plasticity of descending neurons [129], suggesting a role for microglial pruning. Nonetheless, it is still important to specifically assess inhibitory neuron and inhibitory synapse number after nerve injury to disambiguate these two processes; it is likely that both occur simultaneously. This is not an easy task and will require three-dimensional reconstruction with clearing techniques at different time points after injury, or perhaps continuous live imaging of transgenic mice with GFP-labelled inhibitory neurons.

Importantly, synapse specificity of pruning mechanisms has been reported, with increased excitatory and reduced inhibitory synapses [56,72,167,180–183], which would mediate hyperexcitability seen in neuropathic pain. The underlying mechanism for this specificity remains unclear, although it may involve exosome release after hyperexcitability [73], synapse-specific adhesion molecules, heterogeneous action of proteases on specific proteins found at different types of synapses [184], or perhaps differential effects of inhibitory versus excitatory neurotransmitters on microglia [185].

## 5.3. Structural remodelling of excitatory neurons

Microglial secretions and cell-surface interactions can remodel excitatory neurons, as reviewed above (figure 2(3) and (4)). Cytokine secretion by microglia in a neuropathic pain model remodels dendrites in the spinal cord and hippocampus [56]; molecular definition of microglia in different CNS regions would provide mechanistic insight into regional differences, and region-specific chemogenetic activation or inhibition [186] will determine the contribution of localized structural remodelling to the neuropathic pain phenotype. Furthermore, microglia secrete metalloproteinases, C1q and glutamate during neuropathic pain [62,141,179], which are known to remodel excitatory neuron structure. Adhesion molecules, such as integrins and neuroligins, are expressed on microglia and neurons [187] and their interactions are known to direct synaptogenesis and branch formation. In a neuropathic pain model, neuroligin and postsynaptic density protein-95 (PSD-95) co-localization increases, suggesting a role in synaptogenesis, and this apparent change in structure correlates with mechanical hypersensitivity [187]. It is thought that remodelling occurs on both nociceptive specific [157] and wide dynamic range [156] projection neurons in the dorsal horn. Recently, a transcriptomic study has shown regulation of the extracellular matrix genes in animals models of neuropathic pain and also an over-representation of single nucleotide polymorphisms in these genes in patients with back pain [188], suggesting that interactions with the extracellular matrix may play a role in directing circuit remodelling. Indeed, microglia are known to remodel the extracellular matrix around synapses [50], and this may be the first step in synaptic remodelling.

It appears that different microglial phenotypes can mediate synaptogenic and trophic functions, as pro-inflammatory phenotypes activated by zymosan can promote axon growth in an inflammatory pain model [175], whereas CX3CR1-deficient microglia, which express fewer genes involved in inflammatory signalling, appear to show increased synaptogenic and trophic activity [129]. Interestingly, other pro-inflammatory phenotypes appeared unable to mediate these synaptogenic and trophic functions [175], consistent with the known spectrum of microglial phenotypes. It may be that these different phenotypes use different mechanisms as pro-inflammatory phenotypes can secrete glutamate and proteases, whereas trophic phenotypes secrete neurotrophic factors and anti-inflammatory cytokines [129], all of which are implicated in synapse formation and maturation. Although both phenotypes can remodel neural circuitry, it appears that trophic phenotypes promote resolution of symptoms whereas pro-inflammatory phenotypes promote the maintenance of symptoms [129], which suggests that the manner in which they remodel circuitry has different consequences on pain processing.

## 5.4. Ectopic sprouting

Although previous claims for ectopic sprouting of primary afferent input in the dorsal horn have been found to be erroneous, there is still potential for sprouting of primary afferents into the adjacent un-injured dorsal horn regions surrounding the injured afferents. Interestingly, microglial reaction to nerve injury has been demonstrated to spread beyond the edges of injured afferent input [13], and may promote the formation of such sprouting (figure 2(5)).

## 5.5. Neurogenesis and circuit integration

Although not a classical site of adult neurogenesis, a study in 2012 identified mechanosensation-induced neurogenesis in the dorsal horn, resulting in stimulus-specific consolidation and differentiation into mature spinal circuitry [145]. Given this, it is likely that a neuropathic pain stimulus would also affect neurogenesis occurring in the spinal cord, and may alter spinal circuitry to promote a hyperexcitable state or perhaps may mediate rewiring to bring about allodynia (figure 2(6)). This phenomenon could explain the discrepancies in the literature regarding cell death and changes in neuron number after neuropathic pain protocols. It may also explain the presence of immature glycinergic synapses [143] seen in neuropathic pain, which may occur in the process of cell death or circuit integration. It will be interesting to see whether the known role for microglia in remodelling of synaptic proteins, scaffolding proteins and receptor types [55,77] and guidance of maturing synapses [60,66] is carried out in the spinal cord to assist in this rewiring process (figure 2(3)). *In vivo* imaging of inhibitory synapses and microglia will allow assessment of co-localization, pruning and synaptogenesis and how this changes after chemogenetic inhibition of microglia [186] or inducible CSF1 deletion [20]. Further, retroviral transfection for real-time imaging of neurogenesis and cell death would move away from

the static picture provided by staining sections at discrete intervals to a more dynamic picture of spinal circuitry changes in neuropathic pain.

More recent studies have now shown an increase in neuronal nuclei (NeuN[+]) staining and incorporation of 5-ethynyl-2′-deoxyuridine (EdU) into neurons, to quantify replication, after chronic constriction injury [189]. Importantly, pharmacological modulation of neurogenesis allowed bidirectional manipulation of long-term pain thresholds [189], suggesting that integration of immature neurons into spinal circuitry may facilitate maladaptive rewiring and neuropathic pain behaviours. Curiously, they showed that administration of BDNF prevented hyperalgesia by promoting neuronal differentiation of GABAergic neurons which contradicts work with BDNF knock-out mice [11] suggesting a role for BDNF in the chronicity of neuropathic pain. This discrepancy could arise from the fact that BDNF is derived from both microglia and primary afferents; the spatial separation and perhaps different production kinetics may allow each source to mediate different effects. One may direct differentiation of immature neurons into spinal circuits while the other has more of a structural remodelling and disinhibitory role. It is also important to note that while intrathecal BDNF appears to prevent the maintenance of neuropathic pain, this does not exclude the possibility that a more nuanced, controlled and local secretion of BDNF, perhaps by microglia, may direct neuronal differentiation and spinal circuit rewiring to promote hyperexcitability which is not seen by a global, and perhaps supra-physiological, increase in BDNF levels. Alternatively, it may be that the different actions of BDNF occur over different time scales and that pro-nociceptive effects via KCC2 expression [35] and dendritic spine remodelling [56] take place earlier than effects on neuronal differentiation. Elucidating the mechanisms underlying the apparent opposing actions of BDNF on neuropathic pain behaviours and investigating other regulators of neuronal differentiation may present therapeutic targets for neuropathic pain. The same group have also shown structural alterations in the hindbrain and cerebellum after chronic constriction injury which was not seen in Notch3 knock-outs with impaired neuronal differentiation [190], suggesting that this rewiring mechanism occurs at several levels of the somatosensory system.

## 5.6. Remodelling of supraspinal circuitry

Conceptually, remodelling of the pain signature, reward, salience recognition, executive and descending modulatory networks probably plays a significant role in transmission and perception of pain as well as the associated cognitive deficits in memory, concentration and motivation (reviewed in [191]). Microglia are activated in these supraspinal regions in neuropathic pain models [14,15] and in patients [16], and supraspinal remodelling has been shown to occur [161,162]. Microglia have also been shown to modulate supraspinal activity in neuropathic pain models through effects on dopamine release in the ventral tegmental area [192] and on the modulatory activity of the periaqueductal gray [193]. It is likely that the spatial segregation of microglia allows different phenotypes in different regions to differentially modulate the pain signature network; harvesting microglia from biopsies of different brain regions in neuropathic pain models will elucidate the spatial distribution of the molecular signature of microglia at different time points and how this correlates with structural alterations and whole-brain connectivity [194]. Live imaging with *in vivo* two-photon imaging to visualize dendritic spine morphology of cortical neurons in transgenic mice will allow assessment of the changes in structure and $Ca^{2+}$ levels after contact and whether this is affected by chemogenetic inhibition of microglia and inducible conditional deletion of complement and BDNF genes from microglia to identify the underlying mechanisms.

# 6. Conclusion and future perspectives

The framework proposed suggests a pro-inflammatory microglial phenotype modulates the excitability of sensory neurons in the dorsal horn to mediate hyperalgesia. This hyperalgesia is then maintained by structural remodelling, such as pruning and apoptosis of inhibitory neurons, driving disinhibition, and synaptogenesis, resulting in increased spine density of excitatory neurons. The overall result is hyperexcitability and pain memory. However, this is a simplification as no single microglial phenotype is present at one time, but rather a spectrum of phenotypes which mediate similar and complementary functions, and this spectrum may change over time [12,110]. We propose that a more trophic phenotype may promote resolution of symptoms, whereas a more inflammatory phenotype mediates chronicity of neuropathic pain symptoms through maladaptive long-term structural

alterations. In supraspinal regions, distinct microglial phenotypes [56] may remodel connectivity for affective, contextual and modulatory processing of neuropathic pain [192,193] and its associated cognitive symptoms [56]. Microglia also probably play an indirect role in structural remodelling by manipulating activity with secreted molecules [35,124,141,195,196], which drives structural alterations [197]. Other glial cells, such as astrocytes, probably have interdependent but also independent actions to promote remodelling [106,198–201].

Future studies must consider spatial heterogeneity in phenotype [22] and activation [201,202], perhaps due to different neurotransmitter effects [203], and heterogeneity in time of day [185], age [127,128,204] and sex [11]. Furthermore, interactions between cells and molecules can produce distinct effects than when studied in isolation [112] and interactions between systems such as the gut-brain [204] and brain-immune [205] axes probably affect experimental and clinical outcomes.

Combined with improved diagnostic methods [206,207], patient categorization into predicted treatment efficacy based on biomarkers for mechanistic insight [208] (reviewed in [209]), and improved clinical testing based on neural correlates of relief [210] and psychosocial outcome measures, this paradigm may eventually inform clinical practice to alleviate the huge societal burden of neuropathic pain.

Ethics. This is not relevant to this work.

Data accessibility. This article has no additional data.

Authors' contributions. Both authors conducted a review of the literature, formed the general structure for the review and wrote the paper. Both authors gave final approval for publication.

Competing interests. We declare we have no competing interests.

Funding. S.J.W. was previously funded under the Wellcome Strategic Award 'Defining pain circuitry in health and disease ref. 102645', which contributed to this research, and is currently funded by the International Brain Laboratory, Wellcome ref. 216324.

Acknowledgements. We wish to acknowledge the Wellcome, and Dr David Menassa for productive discussions concerning microglial physiology.

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
