## [Reviewer comments · Royal Society Open Science]

Review History

RSOS-200260.R0 (Original submission)

Review form: Reviewer 1

Is the manuscript scientifically sound in its present form?

Yes

Are the interpretations and conclusions justified by the results?

Yes

Is the language acceptable?

No

Do you have any ethical concerns with this paper?

Yes

Have you any concerns about statistical analyses in this paper?

No

Recommendation?

Major revision is needed (please make suggestions in comments)

Comments to the Author(s)

The present review brings an overview of microglia as a population of cells participating in many processes in the central nervous system such as: synapses pruning, synaptogenesis and maintenance of neuron survival. All of those topics are of extreme relevance to understand the role of microglia in a context of disease, both in neurodegenerative and neuropathic disorders. In this manner, the authors seek to correlationate these subjects, focusing on the remodelling functions of this cell population modulating neurons on the course of neuropathic pain.

However, some major issues have to be addressed:

1) The language should be clear and the information discussed should be presented in a more logical way. The introduction drives the reader towards a direction where the microglia could be modulating neurons structure and, therefore, participating in the pain process, but as the manuscript evolves, the reader misses the link between topics, making it difficult to follow the rationale and the main focus of the review.

2) The text has plenty of abbreviations that are not previously explained, thus difficulting the understanding of readers which are just starting to read about this subject. In addition, most of the abbreviations are not contextualized within the topic, which makes it difficult even for experienced readers on the matter.

3) To exemplify the concerns above (topics 1 and 2), here is an extract from the article:
 "A direct role for microglia in synaptogenesis was shown by tamoxifen-inducible CX3CR1 promoter-driven diphtheria toxin expression to selectively ablate microglia resulting in reduced learning-dependent spine turnover and behaviour⁵⁶. Cre-dependent removal of floxed BDNF decreased spine formation and phenocopied fear response behaviour but not novel object behaviour⁵⁶, suggesting that BDNF is only one of the synaptogenic mechanisms employed by microglia. Indeed, recombinant IL-10 mimics, whereas shRNA knock-down prevents, synaptogenesis seen in separated co-culture of hippocampal neurons and microglia⁵⁵. Importantly, neurotrophins can selectively organise regenerative growth allowing specific action despite nonspecific release⁵⁷. However, pre-treatment with LPS prevents synaptogenesis by IL-1 β -mediated inhibition⁵⁵ which also abrogates BDNF-dependent actin-mediated spine consolidation⁵⁸. Therefore, only certain phenotypes of microglia in specific environments are synaptogenic; indeed, after spinal cord injury microglia upregulate repulsive guidance molecule to prevent neurite outgrowth and survival⁵⁹." (Page 5, lines 13-29).

This is the first paragraph of the microglia promoting synaptogenesis topic, so the authors should've explained, briefly, what is the synaptogenesis process and what role the microglia assumes in it in homeostasis. Instead, they choose to start with different results emphasizing the deleterious role of microglia absence or lack of function, not contextualizing it with the original aim of the topic, leaving more questions than answers about it.

Besides that, the scientific language is confusing, as in line 17: "Cre-dependent removal of floxed BDNF decreased spine formation and (...)", where it is not clear if BDNF is being removed from microglia or not.

The information provided is vast and poorly explored. There are missing links between sentences, and it's not possible to construct a logical argument on the subject. Also, the vocabulary is vague: in lines 17-18, for example, the authors use the word "behaviour", but do not delve further in which behaviours could be altered, making it very general. As said previously, many abbreviations and terms are being used (such as: BDNF, neurotrophins, LPS) without being explained the significance of those in the context proposed by the review.

This is just an example of something that occurs through all the manuscript, so the recommendation is to review and correct all those points that could leave margin for confusion.

4) About the figures: there are no subtitles to explain them.

Figure 1 is not mentioned in the manuscript and it has plenty of information but none of them is clear: what's the point of the figure? Where are the receptors being expressed? The

microglial cell is communicating with what, since the drawing is not captioned? We strongly suggest that the image should be re-done, with less information, focusing on less topics.

Figure 2 should be divided in smaller and clearer parts. As it is, it's not possible to understand the participation of microglia in the sculpting of pain pathway nor the proposed techniques to study this phenomenon. This figure also presents conceptual mistakes: the literature show that B cells, T cells and monocytes do not infiltrate in the parenchyma of the spinal cord and there are no previous results that show the secretion of IL-10 in the parenchyma in a context of neuropathic pain (Gu et al. Spinal microgliosis due to resident microglial proliferation is required for pain hypersensitivity after peripheral nerve injury. *Cell Rep.* 2016; Kobayashi et al. TREM2/DAP12 signal elicits proinflammatory response in microglia and exacerbates neuropathic pain. *J Neurosci.* 2016; Guimaraes et al. Blood-circulating leukocytes fail to infiltrate the spinal cord parenchyma after spared nerve injury. *JLB* 2019).

5) Finally, the review proposes an emphasis on microglia and its role during neuropathic pain, suggesting that this population acts as modulators of neuron activity. However, the topic focusing on neuropathic pain is poorly discussed. It is suggested that the subject should be introduced first (what is neuropathic pain and what are the roles already described for microglia in this context) and then the author could make the correlation between the pathophysiology of pain and the studies they cited, which in the way it is written, are out of context and do not permit a complete understanding of the information presented.

Review form: Reviewer 2

Is the manuscript scientifically sound in its present form?

No

Are the interpretations and conclusions justified by the results?

Yes

Is the language acceptable?

Yes

Do you have any ethical concerns with this paper?

No

Have you any concerns about statistical analyses in this paper?

No

Recommendation?

Major revision is needed (please make suggestions in comments)

Comments to the Author(s)

Please see file attached (Appendix A).

Decision letter (RSOS-200260.R0)

31-Mar-2020

Dear Dr West,

The editors assigned to your paper ("Microglia: Sculptors of Neuropathic Pain?") have now received comments from reviewers. We would like you to revise your paper in accordance with the referee and Associate Editor suggestions which can be found below (not including confidential reports to the Editor). Please note this decision does not guarantee eventual acceptance.

Please submit a copy of your revised paper before 23-Apr-2020. Please note that the revision deadline will expire at 00.00am on this date. If we do not hear from you within this time then it will be assumed that the paper has been withdrawn. In exceptional circumstances, extensions may be possible if agreed with the Editorial Office in advance. We do not allow multiple rounds of revision so we urge you to make every effort to fully address all of the comments at this stage. If deemed necessary by the Editors, your manuscript will be sent back to one or more of the original reviewers for assessment. If the original reviewers are not available, we may invite new reviewers.

- Data accessibility

<http://datadryad.org/submit?journalID=RSOS&manu=RSOS-200260>

- Competing interests

- Authors' contributions

- Acknowledgements

- Funding statement

on behalf of Dr Robson da Costa (Associate Editor) and Malcolm White (Subject Editor)
openscience@royalsociety.org

Comments to Author:

Reviewers' Comments to Author:

Reviewer: 1

Comments to the Author(s)

The present review brings an overview of microglia as a population of cells participating in many processes in the central nervous system such as: synapses pruning, synaptogenesis and maintenance of neuron survival. All of those topics are of extreme relevance to understand the role of microglia in a context of disease, both in neurodegenerative and neuropathic disorders. In this manner, the authors seek to correlationate these subjects, focusing on the remodelling functions of this cell population modulating neurons on the course of neuropathic pain.

However, some major issues have to be addressed:

1) The language should be clear and the information discussed should be presented in a more logical way. The introduction drives the reader towards a direction where the microglia could be modulating neurons structure and, therefore, participating in the pain process, but as

the manuscript evolves, the reader misses the link between topics, making it difficult to follow the rationale and the main focus of the review.

2) The text has plenty of abbreviations that are not previously explained, thus difficulting the understanding of readers which are just starting to read about this subject. In addition, most of the abbreviations are not contextualized within the topic, which makes it difficult even for experienced readers on the matter.

3) To exemplify the concerns above (topics 1 and 2), here is an extract from the article:
 "A direct role for microglia in synaptogenesis was shown by tamoxifen-inducible CX3CR1 promoter-driven diphtheria toxin expression to selectively ablate microglia resulting in reduced learning-dependent spine turnover and behaviour⁵⁶. Cre-dependent removal of floxed BDNF decreased spine formation and phenocopied fear response behaviour but not novel object behaviour⁵⁶, suggesting that BDNF is only one of the synaptogenic mechanisms employed by microglia. Indeed, recombinant IL-10 mimics, whereas shRNA knock-down prevents, synaptogenesis seen in separated co-culture of hippocampal neurons and microglia⁵⁵. Importantly, neurotrophins can selectively organise regenerative growth allowing specific action despite nonspecific release⁵⁷. However, pre-treatment with LPS prevents synaptogenesis by IL-1 β -mediated inhibition⁵⁵ which also abrogates BDNF-dependent actin-mediated spine consolidation⁵⁸. Therefore, only certain phenotypes of microglia in specific environments are synaptogenic; indeed, after spinal cord injury microglia upregulate repulsive guidance molecule to prevent neurite outgrowth and survival⁵⁹." (Page 5, lines 13-29).

This is the first paragraph of the microglia promoting synaptogenesis topic, so the authors should've explained, briefly, what is the synaptogenesis process and what role the microglia assumes in it in homeostasis. Instead, they choose to start with different results emphasizing the deleterious role of microglia absence or lack of function, not contextualizing it with the original aim of the topic, leaving more questions than answers about it.

Besides that, the scientific language is confusing, as in line 17: "Cre-dependent removal of floxed BDNF decreased spine formation and (...)", where it is not clear if BDNF is being removed from microglia or not.

The information provided is vast and poorly explored. There are missing links between sentences, and it's not possible to construct a logical argument on the subject. Also, the vocabulary is vague: in lines 17-18, for example, the authors use the word "behaviour", but do not delve further in which behaviours could be altered, making it very general. As said previously, many abbreviations and terms are being used (such as: BDNF, neurotrophins, LPS) without being explained the significance of those in the context proposed by the review.

This is just an example of something that occurs through all the manuscript, so the recommendation is to review and correct all those points that could leave margin for confusion.

4) About the figures: there are no subtitles to explain them.

Figure 1 is not mentioned in the manuscript and it has plenty of information but none of them is clear: what's the point of the figure? Where are the receptors being expressed? The microglial cell is communicating with what, since the drawing is not captioned? We strongly suggest that the image should be re-done, with less information, focusing on less topics.

Figure 2 should be divided in smaller and clearer parts. As it is, it's not possible to understand the participation of microglia in the sculpting of pain pathway nor the proposed techniques to study this phenomenon. This figure also presents conceptual mistakes: the literature show that B cells, T cells and monocytes do not infiltrate in the parenchyma of the spinal cord and there are no previous results that show the secretion of IL-10 in the parenchyma in a context of neuropathic pain (Gu et al. Spinal microgliosis due to resident microglial proliferation is required for pain hypersensitivity after peripheral nerve injury. *Cell Rep.* 2016; Kobayashi et al. TREM2/DAP12 signal elicits proinflammatory response in microglia and exacerbates neuropathic pain. *J Neurosci.* 2016; Guimaraes et al. Blood-circulating leukocytes fail to infiltrate the spinal cord parenchyma after spared nerve injury. *JLB* 2019).

5) Finally, the review proposes an emphasis on microglia and its role during neuropathic

pain, suggesting that this population acts as modulators of neuron activity. However, the topic focusing on neuropathic pain is poorly discussed. It is suggested that the subject should be introduced first (what is neuropathic pain and what are the roles already described for microglia in this context) and then the author could make the correlation between the pathophysiology of pain and the studies they cited, which in the way it is written, are out of context and do not permit a complete understanding of the information presented.

Reviewer: 2

Comments to the Author(s)
Please see file attached.

Author's Response to Decision Letter for (RSOS-200260.R0)

See Appendix B.

RSOS-200260.R1 (Revision)

Review form: Reviewer 1

Is the manuscript scientifically sound in its present form?

Yes

Are the interpretations and conclusions justified by the results?

Yes

Is the language acceptable?

Yes

Do you have any ethical concerns with this paper?

No

Have you any concerns about statistical analyses in this paper?

No

Recommendation?

Accept as is

Comments to the Author(s)

No further comments

Review form: Reviewer 2

Is the manuscript scientifically sound in its present form?

Yes

Are the interpretations and conclusions justified by the results?

Yes

Is the language acceptable?

Yes

Do you have any ethical concerns with this paper?

No

Have you any concerns about statistical analyses in this paper?

No

Recommendation?

Accept with minor revision (please list in comments)

Comments to the Author(s)

The authors have modified and improved the manuscript as requested. However, a few things should be corrected:

- 1) explain the meaning of WAVE1, as it is mentioned only on time in the text and even that it is lacking information about it.
- 2) TNF Knockout model it is not right as a knockout animal is not a model. Thus, it should be something like: a neuropathic pain model induced in TNF knockout mice. The same is mentioned several times in the manuscript, as CXCR3R1 knockout model, among others cytokines knockouts.
- 3) The authors do not explain well what is pruning. Please clarify.
- 4) The titles, subtitles are with numbers in the wrong sequence. For example, page 13 has a topic with number 4, and the following is number 1; which I think it is 4.1. Please revise very careful about the structure of the manuscript.

Decision letter (RSOS-200260.R1)

18-May-2020

Dear Dr West:

On behalf of the Editors, I am pleased to inform you that your Manuscript RSOS-200260.R1 entitled "Microglia: Sculptors of Neuropathic Pain?" has been accepted for publication in Royal Society Open Science subject to minor revision in accordance with the referee suggestions. Please find the referees' comments at the end of this email.

The reviewers and Subject Editor have recommended publication, but also suggest some minor revisions to your manuscript. Therefore, I invite you to respond to the comments and revise your manuscript.

- Ethics statement

- Data accessibility

It is a condition of publication that all supporting data are made available either as supplementary information or preferably in a suitable permanent repository. The data

accessibility section should state where the article's supporting data can be accessed. This section should also include details, where possible of where to access other relevant research materials such as statistical tools, protocols, software etc can be accessed. If the data has been deposited in an external repository this section should list the database, accession number and link to the DOI for all data from the article that has been made publicly available. Data sets that have been deposited in an external repository and have a DOI should also be appropriately cited in the manuscript and included in the reference list.

If you wish to submit your supporting data or code to Dryad (<http://datadryad.org/>), or modify your current submission to dryad, please use the following link:
<http://datadryad.org/submit?journalID=RSOS&manu=RSOS-200260.R1>

- **Competing interests**

- **Authors' contributions**

- **Acknowledgements**

- **Funding statement**

Because the schedule for publication is very tight, it is a condition of publication that you submit the revised version of your manuscript before 27-May-2020. Please note that the revision deadline will expire at 00.00am on this date. If you do not think you will be able to meet this date please let me know immediately.

on behalf of Dr Robson da Costa (Associate Editor) and Malcolm White (Subject Editor)
openscience@royalsociety.org

Associate Editor Comments to Author (Dr Robson da Costa):
Comments to the Author:

The authors have addressed most of the points raised by the reviewers; however, there are still few issues to be addressed, as requested by one of the reviewers, and the MS needs some minor edits.

- 1) To facilitate the readers understanding, the sections of the review should be numbered sequentially, starting from the "Introduction" and finishing in the "Conclusions and Future Perspective". Sub-sections should also be numbered sequentially – e.g. 2. Microglia as Sculptors, 2.1 Microglia as Surveillants... etc... The numbering of the sub-sections in section 4 is confusing; is it correct? Please, make it clearer.

Reviewer comments to Author:

Reviewer: 1

Comments to the Author(s)

No further comments

Reviewer: 2

Comments to the Author(s)

The authors have modified and improved the manuscript as requested. However, a few things should be corrected:

- 1) explain the meaning of WAVE1, as it is mentioned only on time in the text and even that it is lacking information about it.
- 2) TNF Knockout model it is not right as a knockout animal is not a model. Thus, it should be something like: a neuropathic pain model induced in TNF knockout mice. The same is mentioned several times in the manuscript, as CXC3R1 knockout model, among others cytokines knockouts.
- 3) The authors do not explain well what is pruning. Please clarify.
- 4) The titles, subtitles are with numbers in the wrong sequence. For example, page 13 has a topic with number 4, and the following is number 1; which I think it is 4.1. Please revise very careful about the structure of the manuscript.

Author's Response to Decision Letter for (RSOS-200260.R1)

See Appendix C.

Decision letter (RSOS-200260.R2)

01-Jun-2020

Dear Dr West,

It is a pleasure to accept your manuscript entitled "Microglia: Sculptors of Neuropathic Pain?" in its current form for publication in Royal Society Open Science.

Please ensure that you send to the editorial office an individual file for the table included in your manuscript.

on behalf of Dr Robson da Costa (Associate Editor) and Malcolm White (Subject Editor)
openscience@royalsociety.org

Follow Royal Society Publishing on Twitter: [@RSocPublishing](https://twitter.com/RSocPublishing)
Follow Royal Society Publishing on Facebook:
<https://www.facebook.com/RoyalSocietyPublishing.FanPage/>
Read Royal Society Publishing's blog: <https://blogs.royalsociety.org/publishing/>

Appendix A

RSOS-200260

Title: **Microglia: Sculptors of Neuropathic Pain?**

Ward & West review the limitations in experimental design and data analysis published in the literature that investigated the microglia (mainly in the CNS) correlation with neuropathic pain. The authors suggest that microglia induce structural modifications to drive the chronification of neuropathic pain. They also suggest new studies to investigate this mechanism, using more advanced techniques to validate the microglial activation in different regions of CNS and the maintenance of neuropathic pain. In general, the review is not clear, which makes the reading difficult. In addition, some concepts are wander off-topic. The tables and schematic figures are good, but the readers would gain more information if the manuscript was focused on key concepts, mainly the correlation with neuropathic pain that is lacking in several topics.

Comments to the Authors

- 1- The authors should include links between topics and subtopics, as the reading does not flow well. In addition, the topics sequence is not clear, which can be improved with linking paragraphs or sentences.
- 2- The review focus is microglia and neuropathic pain. However, the authors have not included a general topic about microglia, as cell type, expression, among others. Also, evidences about their importance in neuropathic pain which will be the link for a next topic about neuropathic pain. After that, the authors should start the topics about microglia as sculptors.
- 3- An abbreviation list should be included.
- 4- The authors also have to include the meaning of, for example: CX3CR1, P2Y, etc. What kind of receptors are they? What is the correlation with microglia? What is pruning? They cannot suppose the readers will know all the concepts, meanings, mechanisms about microglia. These information's have to be clear, since a review where the readers have to stop the reading to find information that should be present in the paper, it is not interesting.

- 5- It is well known in the literature that P2X receptors are also important. Please include some information about them and microglia and neuropathic pain.
- 6- In several subtopics, the authors explore the microglia as surveillants, actively prune synapses, but none information about their role in neuropathic pain (which is the review topic) is reported. Please review.

The review has to be re-write to make clear the main objective, since in the conformation which is present now, appears only a description of a lot of different ideas without link between them. The review should clarify these ideas and not the opposite.

Appendix B

Dear Andrew Dunn, Dr Robson da Costa, and Malcolm White,

Thank you for considering our review titled ("Microglia: Sculptors of Neuropathic Pain?") and for submitting this for peer review in your journal Royal Society Open Science.

We have carefully considered the reviewers comments, and have comprehensively re-written this review article to fulfil them. Specifically, the structure of the review has become much more defined, which has allowed our argument to build up more logically and clearly. We start with a general overview of microglial physiology, and then discuss the mechanisms of microglial sculpting in the CNS. We then discuss how microglia are involved in neuropathic pain and argue that they mediate the maintenance of neuropathic pain symptoms. We review preliminary studies implicating microglia in structural remodelling of neurons after nerve injury and review evidence that structural remodelling underlies the chronicity of neuropathic pain symptoms. We finish with a detailed framework of potential underlying mechanisms utilised by microglia in this structural remodelling process, collating studies from other CNS regions, and preliminary studies of microglial sculpting of neurons in neuropathic pain.

Please see below our replies to each reviewer's comment.

Kind regards,

Harry Ward and Steven West

Comments to Author:

Reviewers' Comments to Author: Reviewer: 1

Comments to the Author(s)

The present review brings an overview of microglia as a population of cells participating in many processes in the central nervous system such as: synapses pruning, synaptogenesis and maintenance of neuron survival. All of those topics are of extreme relevance to understand the role of microglia in a context of disease, both in neurodegenerative and neuropathic disorders. In this manner, the authors seek to correlationate these subjects, focusing on the remodelling functions of this cell population modulating neurons on the course of neuropathic pain.

However, some major issues have to be addressed:

- 1) The language should be clear and the information discussed should be presented in a more logical way. The introduction drives the reader towards a direction where the microglia could be modulating neurons structure and, therefore, participating in the pain process, but as the manuscript evolves, the reader misses the link between topics, making it difficult to follow the rationale and the main focus of the review.

Thank you for your comment. We agree the language was unclear in places, and we have extensively re-written the review, and many sentences and paragraphs where the intention may have

been unclear have now been adjusted to make the purpose clearer. The structure we outlined in our review proposal has been used as a framework for shaping the review article, and we believe its intention and purpose is now much clearer. In this light, we have also changed the tables. We have removed both tables from the original text as we decided that they were not directly relevant to our main review question. We have instead added in a table discussing key papers looking at the mechanisms of microglial sculpting which is much more in keeping with the rest of the review.

2) The text has plenty of abbreviations that are not previously explained, thus difficulting the understanding of readers which are just starting to read about this subject. In addition, most of the abbreviations are not contextualized within the topic, which makes it difficult even for experienced readers on the matter.

Thank you for your comment. We agree there were shortfalls in our definition of abbreviations, and have ensured each abbreviation is properly defined throughout the text. We have also provided a list of abbreviations that used more than once

3) To exemplify the concerns above (topics 1 and 2), here is an extract from the article:

"A direct role for microglia in synaptogenesis was shown by tamoxifen-inducible CX3CR1 promoter-driven diphtheria toxin expression to selectively ablate microglia resulting in reduced learning-dependent spine turnover and behaviour⁵⁶. Cre-dependent removal of floxed BDNF decreased spine formation and phenocopied fear response behaviour but not novel object behaviour⁵⁶, suggesting that BDNF is only one of the synaptogenic mechanisms employed by microglia. Indeed, recombinant IL-10 mimics, whereas shRNA knock-down prevents, synaptogenesis seen in separated co-culture of hippocampal neurons and microglia⁵⁵. Importantly, neurotrophins can selectively organise regenerative growth allowing specific action despite nonspecific release⁵⁷. However, pre-treatment with LPS prevents synaptogenesis by IL-1 β -mediated inhibition⁵⁵ which also abrogates BDNF-dependent actin-mediated spine consolidation⁵⁸. Therefore, only certain phenotypes of microglia in specific environments are synaptogenic; indeed, after spinal cord injury microglia upregulate repulsive guidance molecule to prevent neurite outgrowth and survival⁵⁹." (Page 5, lines 13-29).

This is the first paragraph of the microglia promoting synaptogenesis topic, so the authors should've explained, briefly, what is the synaptogenesis process and what role the microglia assumes in it in homeostasis. Instead, they choose to start with different results emphasizing the deleterious role of microglia absence or lack of function, not contextualizing it with the original aim of the topic, leaving more questions than answers about it.

Thank you for your comment. We have now extensively edited this section as well as added clearer sections introducing microglia and their general physiology, and the review now leads into the microglial synaptogenesis topic with a brief definition of synaptogenesis.

Besides that, the scientific language is confusing, as in line 17: "Cre-dependent removal of floxed BDNF decreased spine formation and (...)", where it is not clear if BDNF is being removed from microglia or not.

The information provided is vast and poorly explored. There are missing links between sentences, and it's not possible to construct a logical argument on the subject. Also, the vocabulary is vague: in lines 17-18, for example, the authors use the word "behaviour", but do not delve further in which behaviours could be altered, making it very general. As said previously, many abbreviations and terms are being used (such as: BDNF, neurotrophins, LPS) without being explained the significance of those in the context proposed by the review.

This is just an example of something that occurs through all the manuscript, so the recommendation is to review and correct all those points that could leave margin for confusion.

Thank you for your comment. We agree there were areas of confusion in the original draft, and we hope the new version, which has a much clearer structure, has resolved these concerns.

4) About the figures: there are no subtitles to explain them.

Figure 1 is not mentioned in the manuscript and it has plenty of information but none of them is clear: what's the point of the figure? Where are the receptors being expressed? The microglial cell is communicating with what, since the drawing is not captioned? We strongly suggest that the image should be re-done, with less information, focusing on less topics.

Figure 2 should be divided in smaller and clearer parts. As it is, it's not possible to understand the participation of microglia in the sculpting of pain pathway nor the proposed techniques to study this phenomenon. This figure also presents conceptual mistakes: the literature show that B cells, T cells and monocytes do not infiltrate in the parenchyma of the spinal cord and there are no previous results that show the secretion of IL-10 in the parenchyma in a context of neuropathic pain (Gu et al. Spinal microgliosis due to resident microglial proliferation is required for pain hypersensitivity after peripheral nerve injury. *Cell Rep.* 2016; Kobayashi et al. TREM2/DAP12 signal elicits proinflammatory response in microglia and exacerbates neuropathic pain. *J Neurosci.* 2016; Guimaraes et al. Blood-circulating leukocytes fail to infiltrate the spinal cord parenchyma after spared nerve injury. *JLB* 2019).

Thank you for your comment. We agree the figure legends were not as detailed as they could be, and we have added much more detail, as well as completely re-drawn the figures, which we believe resolves these issues. We have also corrected the IL-10 part of the figure. To clarify, this was originally proposed as a hypothesis, and indeed there is evidence immune cells do invade the dorsal horn for T-Cells (PMIDs: 21134441, 18196515, 16674943, 19923276, 12435469), which may be sex-dependent (PMID: 26120961), although the role for monocytes is less clear, and we appreciate this is a contentious issue. However, to retain the focus of this review on microglial mechanisms, we have omitted this point.

5) Finally, the review proposes an emphasis on microglia and its role during neuropathic pain, suggesting that this population acts as modulators of neuron activity. However, the topic focusing on neuropathic pain is poorly discussed. It is suggested that the subject should be introduced first (what is neuropathic pain and what are the roles already described for microglia in this context) and then the author could make the correlation

between the pathophysiology of pain and the studies they cited, which in the way it is written, are out of context and do not permit a complete understanding of the information presented.

Thank you for your comment. We have now expanded the discussion of neuropathic pain, including its definition according to IASP, a brief review of how neuropathic pain is modelled and studied, and the typical phenotypes that are seen. We have also made clear the contribution of microglia to the neuropathic pain phenotype, fitting with our re-arrangement of the presentation of the material.

Reviewer: 2

Comments to the Author(s)

Ward & West review the limitations in experimental design and data analysis published in the literature that investigated the microglia (mainly in the CNS) correlation with neuropathic pain. The authors suggest that microglia induce structural modifications to drive the chronification of neuropathic pain. They also suggest new studies to investigate this mechanism, using more advanced technics to validate the microglial activation in different regions of CNS and the maintenance of neuropathic pain. In general, the review is not clear, which makes the reading difficult. In addition, some concepts are wander off-topic. The tables and schematic figures are good, but the readers would gain more information if the manuscript was focused on key concepts, mainly the correlation with neuropathic pain that is lacking in several topics.

Comments to the Authors

- **1-** The authors should include links between topics and subtopics, as the reading does not flow well. In addition, the topics sequence is not clear, which can be improved with linking paragraphs or sentences.

Thank you for your comment. We agree the original structure was not very clear, and so we have re-written the review around the structure of our review proposal. We hope this has made the work much clearer to understand.

- **2-** The review focus is microglia and neuropathic pain. However, the authors have not included a general topic about microglia, as cell type, expression, among others. Also, evidences about their importance in neuropathic pain which will be the link for a next topic about neuropathic pain. After that, the authors should start the topics about microglia as sculptors.

Thank you for your comment. We have made clearer and expanded a general introduction on microglial function in the nervous system, and also known contributions of microglia to the neuropathic phenotype, both in clearly marked sections in our revised structure.

- **3-** An abbreviation list should be included.

Thank you for your comment. We have now ensured every abbreviation used is first defined at its first use. We have also provided a list of abbreviations that used more than once.

- **4-** The authors also have to include the meaning of, for example: CX3CR1, P2Y, etc. What kind of receptors are they? What is the correlation with microglia? What is pruning? They cannot suppose the readers will know all the concepts, meanings, mechanisms about microglia. These information's have to be clear, since a review where the readers have to stop the reading to find information that should be present in the paper, it is not interesting.

Thank you for your comment. As part of our re-writing of the introduction to microglial function in the brain, we have included a part on discussing different receptors and their roles in this physiology. We have also ensured we introduce different concepts before delving into the literature, and have organised the material under clear headings to make the structure more digestible.

-
- **5-** It is well known in the literature that P2X receptors are also important. Please include some information about them and microglia and neuropathic pain.

Thank you for your comment. We have now added these important studies implicating P2X receptors on microglia in neuropathic pain.

- **6-** In several subtopics, the authors explore the microglia as surveillants, actively prune synapses, but none information about their role in neuropathic pain (which is the review topic) is reported. Please review.

Thank you for your comment. We have now added a clearly labelled section reviewing the known roles and evidence for microglia in the neuropathic pain phenotype.

The review has to be re-write to make clear the main objective, since in the conformation which is present now, appears only a description of a lot of different ideas without link between them. The review should clarify these ideas and not the opposite.

Thank you for your comment. We have re-written this review using our original review proposal as a framework for structuring it. We now believe the work reads much more clearly, and this helps it to communicate its intended purpose - to develop ideas around the expanding roles for microglia in brain function and how these may potentially contribute to the neuropathic pain phenotype.

Appendix C

Dear Anita Kristiansen, Dr Robson da Costa, and Malcolm White,

Thank you for accepting our review titled ("Microglia: Sculptors of Neuropathic Pain?") for publication subject to minor revision.

We have implemented the minor edits as suggested in the decision letter. Please see below our point-by-point responses to each comment.

Kind regards,

Harry Ward and Steven West

Associate Editor Comments to Author (Dr Robson da Costa):
Comments to the Author:

The authors have addressed most of the points raised by the reviewers; however, there are still few issues to be addressed, as requested by one of the reviewers, and the MS needs some minor edits.

- 1) To facilitate the readers understanding, the sections of the review should be numbered sequentially, starting from the "Introduction" and finishing in the "Conclusions and Future Perspective". Sub-sections should also be numbered sequentially – e.g. 2. Microglia as Sculptors, 2.1 Microglia as Surveillants... etc... The numbering of the sub-sections in section 4 is confusing; is it correct? Please, make it clearer.

Thank you for your comment. We have implemented this numbering system, and clarified the numbering of section 5.

Reviewer comments to Author:

Reviewer: 1

Comments to the Author(s)

No further comments

Reviewer: 2

Comments to the Author(s)

The authors have modified and improved the manuscript as requested. However, a few things should be corrected:

- 1) explain the meaning of WAVE1, as it is mentioned only on time in the text and even that it is lacking information about it.

Thank you for your comment. We have clarified the meaning of WAVE1 and its role in regulating the actin cytoskeleton.

- 2) TNF Knockout model it is not right as a knockout animal is not a model. Thus, it should be something like: a neuropathic pain model induced in TNF knockout mice. The same is mentioned several times in the manuscript, as CX3CR1 knockout model, among others cytokines knockouts.

Thank you for your comment. We have altered this as suggested throughout the text.

3) The authors do not explain well what is pruning. Please clarify.

Thank you for your comment. We have added in a definition of pruning as the removal of synaptic elements classically but not exclusively by engulfment of synaptic terminals through phagocytosis, and referred to Figure 1a which depicts synaptic pruning.

4) The titles, subtitles are with numbers in the wrong sequence. For example, page 13 has a topic with number 4, and the following is number 1; which I think it is 4.1. Please revise very careful about the structure of the manuscript.

Thank you for your comment. We have added in a numbering system to the text with subsections (e.g. 4.1) as suggested to clarify this.